# New Gall-Forming Insect Model, *Smicronyx madaranus*: Critical Stages for Gall Formation, Phylogeny, and Effectiveness of Gene Functional Analysis

**DOI:** 10.3390/insects15010063

**Published:** 2024-01-16

**Authors:** Ryo Ushima, Ryoma Sugimoto, Yota Sano, Hinako Ogi, Ryuichiro Ino, Hiroshi Hayakawa, Keisuke Shimada, Tsutomu Tsuchida

**Affiliations:** 1Graduate School of Science and Engineering for Education, University of Toyama, Toyama City 930-8555, Toyama, Japan; 2School of Science, University of Toyama, Toyama City 930-8555, Toyama, Japan; 3Museum of Natural and Environmental History, Shizuoka, Shizuoka City 422-8017, Shizuoka, Japan; 4Ishikawa Museum of Natural History, Ri-441, Choshi-Machi, Kanazawa City 920-1147, Ishikawa, Japan; 5Faculty of Science, Academic Assembly, University of Toyama, 3190 Gofuku, Toyama City 930-8555, Toyama, Japan

**Keywords:** insect gall, manipulation of galling insect, field dodder, phylogeny of the genus *Smicronyx*, RNA interference

## Abstract

**Simple Summary:**

A new gall-forming model, *Smicronyx madaranus*, showed that gall formation consists of two processes: initiation by adults and enlargement by larvae. Phylogenetic analysis showed that gall-forming weevils belong to two distinct lineages that utilize different host plants, suggesting that gall-forming traits evolved independently in these *Smicronyx* lineages. This study demonstrates that RNAi is effective for functional gene analysis in *S. madaranus*. The *S. madaranus* model contributes to elucidating the mechanisms of gall formation and understanding the commonalities and diversity of insect galls.

**Abstract:**

The molecular mechanisms underlying insect gall formation remain unclear. A major reason for the inability to identify the responsible genes is that only a few systems can be experimentally validated in the laboratory. To overcome these problems, we established a new galling insect model, *Smicronyx madaranus*. Our manipulation experiments using nail polish sealing and insecticide treatment revealed an age-dependent change in gall formation by *S. madaranus*; adult females and larvae are responsible for gall induction and enlargement, respectively. Furthermore, it has been suggested that substances released during oviposition and larval feeding are involved in each process. Phylogenetic analysis showed that gall-forming weevils, including *S. madaranus*, belong to two distinct lineages that utilize different host plants. This may indicate that gall-forming traits evolved independently in these *Smicronyx* lineages. The efficacy of RNA interference (RNAi) in *S. madaranus* was confirmed by targeting the multicopper oxidase 2 gene. It is expected that the mechanisms of gall formation will be elucidated by a comprehensive functional analysis of candidate genes using RNAi and the *S. madaranus* galling system in the near future.

## 1. Introduction

Some insects induce abnormally enlarged tissues called “galls” in plants [1,2]. Galls function as shelters for inducers and protect them from natural enemies and environmental stressors. They also serve as feeding sites for gall-forming insects [3,4]. Therefore, the gall is conceivable as an “extended phenotype” of insects [5]. However, the mechanisms underlying gall formation in insects remain unclear. The phytohormones auxins and cytokinins biosynthesized in insects have been proposed to be important contributors to the process of gall formation [6,7]. In addition, certain protein effectors potentially released by insects are candidates for initiating plant gall formation [8,9,10,11]. However, the functions of potential protein effectors have only been implicated by RNA-seq and genomic analyses, and the causal relationship between candidate genes and gall formation has not been confirmed. A major reason for the inability to identify genes involved in gall formation is that most insect galls are produced on woody plants [12,13] and only a few systems can be experimentally validated in the laboratory [14]. To overcome these problems, a model system for plants and insects that can be manipulated in the laboratory is required.

Most weevils of the genus *Smicronyx* feed on parasitic plants of the genus *Cuscuta*, and some species form galls on plants [15,16]. *Smicronyx madaranus* Kono, 1930 feeds on *C. campestris* and forms spherical fruit-like galls on it [17]. In a previous study, we established a stable maintenance system for *S. madaranus* and *C. campestris* in the laboratory as a new galling model system and investigated their gall-forming behavior, gall formation process, and histochemical and physiological features [18]. The study showed that adult females bored holes in the nodes of *C. campestris* with their long rostrums and laid eggs inside the holes; then, the tissues of the nodes enlarged and transformed into spherical galls (Appendix A). The larvae grew by feeding on the starch-rich tissue inside the gall and emerged in approximately 12 days. Notably, *S. madaranus*-induced galls greatly increase the photosynthetic activity of the holoparasitic plant *C. campestris*, which depends on the host plant for its nutrients and shows low photosynthetic activity [18]. This is a specific feature of *S. madaranus*-induced galls, in contrast to the general galls formed on the leaves of various plants in which photosynthetic gene expression is suppressed [19,20,21]. This implies that general plant galls undergo a transformation from a nutrient-producing organ (source) to a nutrient-receiving organ (sink), whereas *S. madaranus*-induced galls undergo the opposite transformation. Therefore, it is important to elucidate the mechanism of gall formation in *S. madaranus* to understand the commonality and diversity among insect galls.

In this study, as a preliminary step towards elucidating the molecular mechanisms involved in gall formation, we attempted to clarify the growth stages of the weevil involved in gall induction and enlargement through manipulation experiments on *S. madaranus.* We also clarified the phylogenetic position of *S. madaranus* in relation to other gall-forming and non-gall-forming species. In addition, the efficacy of RNAi in *S. madaranus* as a functional gene analysis method was evaluated by targeting the multicopper oxidase 2 (*MCO2*) gene, a phenotypic marker gene.

## 2. Materials and Methods

### 2.1. Insect and Plant

A laboratory strain of *S. madaranus* was collected from Neagari Nomi City, Ishikawa, Japan. It was maintained on *C. campestris* parasitizing *Nicotiana benthamiana* in the laboratory at 28 °C in a long day regimen (14L10D). *Cuscuta campestris* (with three to five shoots, approximately 10 cm long) parasitizing *Vicia faba* (approximately 25 cm long) was used for some experiments, as described later. Rearing and cultivation conditions were the same as those described previously [17]. To increase survival from the pupal to adult stages, the following modifications were made: Last instar larvae emerging from the galls were collected in a Petri dish S35-DC12 (Fine Plus International, Kyoto, Japan) containing vermiculite moistened with water. The Petri dish was placed in a container containing a saturated saline solution to maintain a humidity of 75%. The entire container was covered with aluminum foil to keep it dark and stored at 28 °C. Individuals that metamorphosed into pupae were transferred to new Petri dishes lined with paper. The Petri dishes were then placed back into the container to complete the pupal-to-adult metamorphosis.

In addition to *S. madaranus*, *S. dentirostris*, *S. japonicus*, and *S. rubricatus* were collected from various locations in central Japan (Appendix A). The samples were preserved in 100% acetone until use in molecular phylogenetic analyses [22].

### 2.2. Observation of S. madaranus in the Initial Gall

Initial galls (1.5–2 mm in width, Figure 1A) appeared approximately 3 days after weevil introduction on *C. campestris* [17]. Twenty initial galls were collected from a laboratory-maintained colony. We then dissected the galls with forceps under an M165C stereomicroscope (Leica, Wetzlar, Germany) and observed the weevils located inside the gall.

### 2.3. Ovipositor Sealing Experiment

It is difficult to distinguish between the sexes of *S. madaranus* based on appearance. Therefore, we identified females for this experiment by observing their mating behavior as follows: Twenty young adults were kept in a small FALCON no. 35100 Petri dish (60 × 15 mm) (Corning, NY, USA) and allowed to mate for 2 days. During this period, five shoots of *C. campestris* cut into 10 cm lengths were placed in the Petri dish as food. Shoots were replaced daily. Individuals on the underside during mating were considered females and used in the experiment. To inhibit oviposition, white nail polish was applied with a toothpick to cover the tail ends of the females (Appendix A) under a stereomicroscope M165C (Leica). In the control group, the same nail polish was applied to the elytra.

To determine whether the sealing treatment affected the locomotor activity, we measured and compared the distance between sealed and control individuals. One day after treatment, weevils were placed in a container made of a silicon O-ring (17.8 mm inner diameter) and cover glass (18 mm × 18 mm, Matsunami Glass Industry, Osaka, Japan) and placed on a white LED light (3.2 mW/m^2^) (Appendix A). Their behavior was then recorded directly for 5 min using a Camera Module v.2.1 connected to a Raspberry Pi 3 Model B (Raspberry Pi Foundation, Cambridge, UK) (Appendix A). The video was analyzed using the original source code written in R language [23] v.4.2.1. The script code detected the points at which the weevils were located at any given moment and calculated the distance they had moved by summing the distances between the points at different times. The experiments were repeated thrice. Ten individuals from each treatment group were used for each experiment.

Ovipositor-sealed and control weevils were individually placed in a box containing *C. campestris* parasitizing *V. faba* seedlings and maintained for four days. Then, 20 nodes of *C. campestris* were cut and collected from the box. The nodes were examined for egg-laying holes and gall formation using a Leica M165C stereomicroscope.

### 2.4. Effects of the Larva on Gall Formation

Ten young adult weevils (of mixed sex) were placed in a box containing three seedlings of *V. faba* parasitized with *C. campestris* and allowed to form galls. All the weevils were removed from the boxes after 48 h. The following two insecticides were then applied to the emerged initial galls to eliminate any larvae present inside; Dantotsu water solvent insecticide (Sumitomo Chemical, Tokyo, Japan) is a neonicotinoid-based agent containing 16.0% clothianidin; Ortolan granules (Sumitomo Chemical) are an organic phosphorus-based agent containing 50.0% acephate. These insecticides have different modes of action; therefore, it is unlikely that both insecticides would directly affect the gall formation mechanisms in *C. campestris*. Both insecticides were diluted with sterile water to 1/1000 (*v*/*v*), and Squash Adhesive [final concentration: 0.1% (*v*/*v*)] (Maruwa Biochemical, Tokyo, Japan) was added to improve penetration into the gall. Two microliters of each insecticide solution were applied to the galls using a micropipette. The control group was treated with 0.1% Squash Adhesive. The galls were marked with markers. To confirm the larval status inside the galls, five days after treatment, the galls were cut with a razor blade, and the interiors were observed using a Leica M165C stereomicroscope. Twenty-four, twenty-one, and seventeen galls were examined in the control, Dantotsu, and Ortholan treatments, respectively. The effect of insecticide treatment on gall enlargement was analyzed as follows. Images of the galls were captured with a ruler every 24 h using a digital camera (Stylus TG-3 Tough; Olympus, Tokyo, Japan). Gall widths were calculated through image analysis using Fiji [24] and compared between the treatment and control groups.

### 2.5. Molecular Phylogenetic Analysis of S. madaranus and Its Related Species

DNA was isolated from samples (Appendix A) using a conventional phenol extraction method. The purified DNA was resuspended in 20 µL of TE buffer [10mM Tris-HCl (pH 8.0), 0.1 mM EDTA]. Partial sequences of cytochrome c oxidase subunit I (COI) were amplified using KOD-FX Neo DNA polymerase (Toyobo, Osaka, Japan) with the primers LCO1490 and HCO2198 (Appendix A). The PCR was carried out with the following temperature profile: 94 °C for 2 min, followed by 35 cycles of 98 °C for 10 s, 50 °C for 30 s, and a final extension at 68 °C for 30 s. The PCR products were purified using a polyethylene glycol solution (PEG6000 20% and 2.5 mM). The PCR amplicons were directly sequenced in both directions with the same primers using a BigDye Terminator v3.1 Cycle Sequencing Kit and Applied Biosystems 3500 Genetic Analyzer (Thermo Fisher Scientific, Waltham, MA, USA). The resulting sequences were assembled and analyzed using ATGC ver. 4 software (GENETYX, Tokyo, Japan).

For the molecular phylogenetic analysis, the COI sequences of four species in Japan and previously published sequences of the genus *Smicronyx* were included in the dataset, and *Tychius capucinus* and *T. breviusculus* were used as outgroups. The dataset included five gall-forming species (*S coecus*, *S. guineanus*, *S. jungermanniae*, *S. madaranus*, and *S. smreczynskii*) [17,25,26,27,28,29] and 18 species (*S. albosquamosus*, *S. australis*, *S. dentirostris*, *S. fallax*, *S. gracillipes*, *S. japonicus*, *S. jordanicus*, *S. longitarsis*, *S. lutulentus*, *S. nebulosus*, *S. pauperculus*, *S. pseudocoecus*, *S. reichi*, *S. rubricatus*, *S. rufus*, *S. san*, *S. syriacus*, and *S. zonatus*) with no description of gall formation [16,17,25,26,30,31,32,33,34,35,36]. Analyses were performed using Bayesian inference (BI), maximum likelihood (ML), and maximum parsimony (MP) criteria. For BI, the most appropriate model of sequence evolution (TN93+G model) was selected using the MEGA X model-selection option [37]. The parameters for the selected substitution model were estimated using the data. A total of 100,000 trees were obtained (ngen = 10,000,000, sample freq = 100) using MrBayes 3.2.7 [38], and the first 25% of these trees (25,000) were considered ‘burned in’ and discarded. A consensus tree was constructed from the remaining trees based on a 50% majority rule. Two independent runs were performed using the same sequence evolution model. For ML, bootstrap analysis of 1000 replications was performed based on the same model as BI in MEGA X. Initial trees for the heuristic search were automatically obtained by applying the Neighbor-Join and BioNJ algorithms to a matrix of pairwise distances estimated using the maximum composite likelihood approach. For MP, all characters were included and weighed equally, and 1000 bootstrap replicates were performed using MEGA X. The Subtree–Pruning–Regrafting algorithm was used with search level 1, where the initial trees were obtained through the random addition of sequences (10 replicates). Gall formation traits were mapped to the resulting phylogenetic tree.

### 2.6. Expression Analysis and RNAi for the S. madaranus Multicopper Oxidase 2 Gene (SmMCO2)

Two candidate *MCO2* genes (hereafter *SmMCO2A* and *SmMCO2B*) were identified in *S. madaranus* through RNA sequencing, which was performed by a collaborator (K. Bessho-Uehara, in preparation), and confirmed through Sanger sequencing (Appendix A). Samples used for gene expression analysis were prepared as follows: Weevils were collected from stages P2 to A3, as defined in this study (Appendix A). Total RNA was extracted from each sample, consisting of three mixed-sex individuals, using the NucleoSpin RNA XS Kit (Takara-Bio, Shiga, Japan). Using 300 ng of total RNA, 20 µL of cDNA was synthesized using the PrimeScriptII; 1st Strand cDNA Synthesis Kit (Takara-Bio) and the Random 6 mer primer supplied with the kit. Quantitative PCR of the *SmMCO2* genes was performed using THUNDERBIRD Next SYBR qPCR Mix (Toyobo) and Mx3005P (Agilent Technologies, Santa Clara, CA, USA) with specific primer sets, as shown in Appendix A. In this study, we analyzed the expression of both *SmMCO2A* and *SmMCO2B* using a single primer set, without distinguishing between them (Appendix A). The PCR temperature profile was 95 °C for 1 min, followed by 40 cycles of 95 °C for 5 s, 62 °C for 30 s, and a final extension at 72 °C for 30 s. The dissociation stage was performed at 95 °C for 15 s, 62 °C for 1 min, followed by a slow ramp to 95 °C. Quantitative PCR and dissociation curve analyses were performed on three samples per stage using a standard curve method, as previously described [39,40].

Double-stranded RNA (dsRNA) targeting the *SmMCO2* genes was synthesized through in vitro transcription of PCR amplicons containing the T7 promoter using RiboMAX Large Scale RNA Production Systems (Promega, Madison, WI, USA). The PCR-generated DNA template was amplified using KOD-Plus-Neo (Toyobo) and the specific primer sets listed in Appendix A. The PCR temperature profile was 95 °C for 2 min, followed by five cycles of 98 °C for 10 s, 58 °C for 30 s, and a final extension at 68 °C for 20 s. This was followed by 30 extended cycles of 98 °C for 10 s, 62 °C for 30 s, and a final extension at 68 °C for 30 s. dsRNA targeting the enhanced green fluorescent protein (EGFP) gene was synthesized as described above and used as a control. A total of 0.5 µg of dsRNA for each gene was injected into last instar larvae (Appendix A) using a capillary needle made from a glass capillary (Drummond Scientific Company, Broomall, PA, USA. 1–5 µL, 90 mm long) with a PN-31 needle puller (Narishige, Tokyo, Japan). Eight days after dsRNA injection, total RNA was extracted from each individual using the Maxwell RSC instrument and Maxwell RSC simplyRNA Tissue Kit (Promega). Quantitative PCR and dissociation curve analysis for the *SmMCO2* genes were performed on 15 samples each from the RNAi treatment and control groups, as described above. We then observed the effect of RNAi on the body color of weevils at 12 and 24 h after adult eclosion.

### 2.7. Nucleotide Sequence Accession Numbers

The sequences reported here have been submitted to the DDBJ/EMBL/GenBank database under the accession numbers LC778253–LC778268 for COI and LC789100 and LC789101 for *SmMCO2A* and *SmMCO2B*, respectively.

### 2.8. Statistics

Behavioral activities and gall formation numbers were compared between the ovipositor-sealed and control groups using the Wilcoxon rank sum test and Fisher’s exact test, respectively. Fisher’s exact test with Bonferroni correction was used to compare survival between the two insecticides and the controls. The pairwise Wilcoxon rank sum test with Bonferroni correction was used to assess the effect of insecticides on the width of the developed gall. One-way analysis of variance (ANOVA) was used to evaluate the differences in *SmMCO2* gene expression levels between growth stages. Tukey’s test was used for post hoc multiple comparisons. Welch’s *t*-test was used to evaluate the differences in *SmMCO2* gene expression levels between the RNAi-treated and control groups. All statistical analyses were performed using R software [23] v.4.2.1.

## 3. Results and Discussion

### 3.1. Galls Are Induced by Adult Females, Not by Larvae

Our previous study showed that galls were induced by weevil oviposition after boring with its long rostrum [18]. When we dissected the initially developed galls (Figure 1A), eggs—not larvae—were always present inside (20/20 galls) (Figure 1B). This finding indicates that gall induction occurs before the larvae hatch. This suggests that gall induction is caused by adult saliva or eggs and the substances delivered with them.

### 3.2. Gall Formation Is Initiated with Substances Delivered during Oviposition

Eggs of herbivorous insects have been reported to induce plant responses [41,42,43]. Eggs and oviposition fluids are commonly involved in gall formation in hymenopteran groups [44,45,46,47] and some weevil species [48,49]. To test the possibility that galls were induced by substances delivered with the eggs of *S. madaranus*, ovipositor sealing experiments were performed (Appendix A). Ovipositor sealing resulted in a slight decrease in movement compared to that in the control group but was not significantly different in all three experiments (Appendix A). Egg-laying holes and galls were formed in the control group (Table 1). The ovipositor-sealed group exhibited fewer egg-laying holes. This may have been due to a decrease in locomotor activity (Appendix A). Although egg-laying holes were observed, no galls formed in the ovipositor-sealed group (Table 1). These results suggest that gall induction is strongly related to eggs and/or substances delivered during oviposition and not to saliva released during egg-laying hole formation.

### 3.3. Gall Enlargement Is Influenced by Larva

Our previous study showed that gall growth stopped and gall size decreased after *S. madaranus* larvae escaped [18]. This suggests that gall growth is strongly influenced by the presence of weevil larvae inside the galls. To test this possibility, we eliminated larvae from the galls using insecticides and examined their influence on gall growth.

First, insecticidal efficacy was assessed five days after treatment. In the control group, surviving larvae were found in almost all the galls (Figure 2A). In contrast, only one surviving larva was found for each of the two insecticide treatments. Survival was significantly different between the control and Dantotsu treatments and between the control and Ortolan treatments. On the other hand, no significant differences were observed between the two insecticide groups (Figure 2A). These results indicated that these insecticides were sufficiently effective in killing *S. madaranus* in galls.

Next, we analyzed the effect of killing larvae on gall enlargement. The control galls increased in size over time. However, when the weevils were killed with either insecticide, the galls stopped enlarging (Figure 2B). These results indicate that the presence of larvae in the galls is responsible for their enlargement. *Smicronyx madaranus* larvae feed vigorously inside the galls, and their inner cells show active cell division, presumably in response to larval ingestion [18]. Therefore, saliva-derived substances produced by *S. madaranus* larvae may be involved in the enlargement of galls.

In previously reported weevil species, both the initiation and enlargement of galls were driven by either adult oviposition (e.g., *Ceutorhynchus napi* on *Brassica napus*; *Rhinusa pilosa* on *Linaria vulgaris*) [42,49] or larval feeding (e.g., *Acythopeus burkhartorum* on *Coccinia grandis*; *Conotrachelus albocinereus* on *Parthenium hysterophorus*) [50,51]. In *S. madaranus*, the two gall-formation processes are differentially controlled by adult oviposition and larval feeding. This suggests that gall-formation mechanisms are diverse, even among different weevil species.

### 3.4. Phylogenetic Analysis of the Smicronyx madaranus and Its Related Species

Phylogenetic analysis of the mitochondrial COI genes was conducted for 23 species of the genus *Smicronyx*, including *S. madaranus*. *Smicronyx smreczynskii* Solari, 1952 was found to be in the same clade as *S. madaranus*. *Smicronyx madaranus* and *S. smreczynskii* are similar in appearance and nature; they form spherical galls on *Cuscuta* plants with increased photosynthetic activity [18,52]. This result suggests that *S. madaranus* and *S. smreczynskii* are synonymous for the same species. However, a detailed analysis is required to test this possibility.

The analysis included three other *Smicronyx* species, *S. dentirostris*, *S. japonicus*, and *S. rubricatus*, which coexist with *S. madaranus* in Japan [16,17,30]. All these species use *Cuscuta* spp. as host plants. In particular, *S. japonicus* shares the same host plant, *C. campestris*, with *S. madaranus*. Despite being found in similar environments, *S. madaranus* was shown to belong to a different clade than the others (Figure 3). The results also indicate that the gall-forming species were divided into two clades. *Smicronyx coecus*, *S. jungermanniae*, *S. madaranus*, and *S. smreczynskii* belong to one clade, whereas *S. guineanus* belongs to a separate clade (Figure 3). Notably, the former four species utilize the genus *Cuscuta* as host plants [18,25,28] similar to many other *Smicronyx* species, whereas the latter one utilizes parasitic plants of the genus *Striga* [26]. These results may indicate that the gall-forming trait has evolved independently in the *Smicronyx* lineages that utilize different host plants.

To date, more than 130 species have been described in the genus *Smicronyx* [16,26,31,34]; however, the species used in this study did not cover the entire genus. In the future, more detailed phylogenetic analyses using high-resolution genetic markers for a larger number of species are required to provide a complete picture of the evolution of gall-forming traits in this genus.

### 3.5. RNAi Is Effective for Gene Silencing in S. madaranus

In recent years, RNA-seq and genome analyses of gall-forming insects have been performed, and several candidate genes associated with gall formation have been identified [9,11,53]. However, functional analysis of the genes is necessary to prove that they are responsible for the gall formation. RNAi is a powerful research tool for gene function analysis but can be inefficient in some insects [54]. Therefore, the efficiency of RNAi is a critical factor in determining its utility as a model for gall-forming insects. To assess the efficacy of RNAi in *S. madaranus*, we knocked down the multicopper oxidase 2 (*SmMCO2*) gene, which is involved in cuticle melanization [55]. This gene was chosen because its knockdown results in a visible change in body color, making it easy to evaluate the efficacy of RNAi.

*SmMCO2* expression was low in P2 (early pupal stage) and P3 (mid-pupal stage) (Figure 4A and Appendix A). The expression level of *SmMCO2* increased rapidly from P4 (late pupal stage), when the black spotted pattern appeared on the wings. The expression then continued to increase until A2 (mid-adult stage), when the head capsule and thorax were melanized. After the A2 stage, *SmMCO2* expression tended to decrease as the whole body became black. We injected dsRNA of *SmMCO2* into the last instar larvae and observed the effect on the gene expression and the body color blacking. dsRNA treatment effectively decreased *SmMCO2* expression (Figure 4B). *SmMCO2*-RNAi individuals did not show any black spot patterns on their wings in the A1 stage (Figure 4C). The body color of the *SmMCO2*-RNAi group remained light brown, even after the whole body of the control group darkened. These results indicate that RNAi is effective for functional gene analysis in *S. madaranus*. In this study, *SmMCO2* was identified as a gene involved in the darkening of body color and the formation of a characteristic spotted pattern on the elytra of *S. madaranus*, as described in the coloration of other insects [55].

## 4. Conclusions and Perspectives

We found that *S. madaranus* is an excellent model for research on the mechanisms of gall formation. It can be maintained throughout the year and subjected to various manipulations in the laboratory, which will help accelerate research.

Our manipulation experiments revealed that there is an age-dependent role change in gall formation in *S. madaranus*; adult females and larvae are responsible for gall induction and enlargement (Table 1, Figure 1 and Figure 2). Substances released during oviposition and larval feeding are expected to be involved in each process. Phytohormones and protein elicitors released by oviposition and feeding have been suggested to play important roles in gall formation [6,7,8,10,11,42,45,49,56,57]. To understand the insect-derived factors involved in gall formation, it will be necessary to identify candidate genes, phytohormones, and other factors in each developmental stage of *S. madaranus*. This can be achieved through future investigations using RNA-seq and LC-MS analyses.

Our phylogenetic analysis showed that the gall-forming weevils, including *S. madaranus*, belong to two distinct lineages that utilize different host plants (Figure 3). This may indicate that the gall-forming trait evolved independently in these *Smicronyx* lineages. Gall-forming weevils are thought to have evolved from seed or fruit feeders [15]. The ability to produce galls, a fruit-like food source, is considered a highly adaptive evolution, as host plants produce flowers only for limited times of the season. Recent molecular biology studies have shown that the expression of flower and fruit development genes is significantly upregulated in the developing galls of several plant species [19,21,58]. Similar mechanisms may be involved in the development of galls in *C. campestris*. Further experimental studies using RNA-seq are required. Comparative genomic analyses between closely related gall-forming and non-gall-forming species will be useful for elucidating the molecular evolution of *Smicronyx* weevils that enable gall formation.

In this study, we showed that RNAi was effective in adult *S. madaranus* through an experiment targeting the *MCO2* gene. It is expected that the mechanisms of gall induction in adult females can be elucidated through a comprehensive functional analysis of candidate genes using RNAi. However, the injection of dsRNA is difficult for larvae living in the gall. Therefore, other techniques are needed for the functional analysis of the genes expressed in larvae. Maternal RNAi [59], feeding-RNAi via the host plant [60], and genome editing [61] will be useful for this purpose.

## Figures and Tables

**Figure 1 insects-15-00063-f001:**
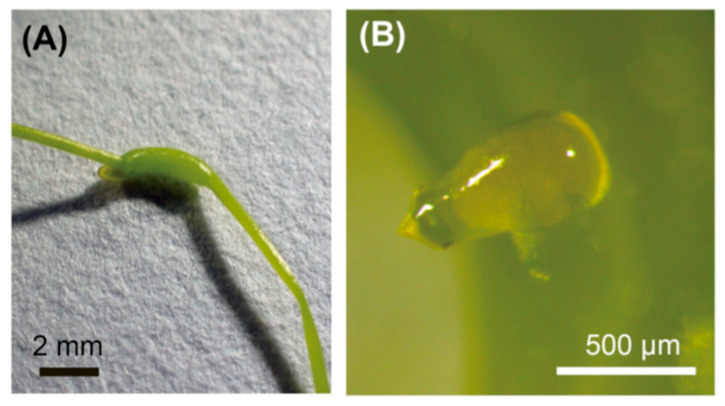
An initial gall (**A**) and an egg in the initial gall (**B**).

**Figure 2 insects-15-00063-f002:**
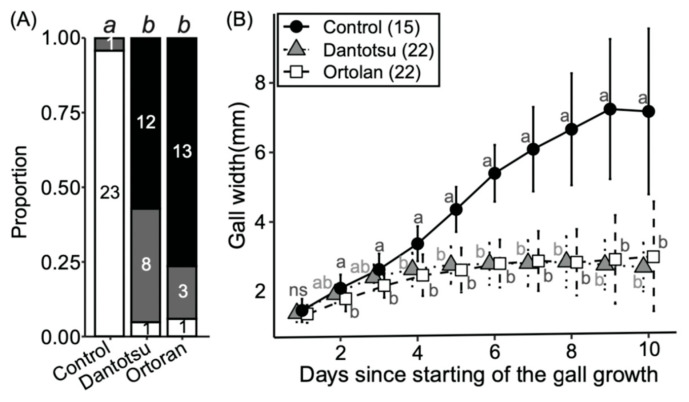
Effects of insecticides on weevil larvae and gall growth. (**A**) Larval status within treated galls. White background, alive; dark grey, undetected; black, dead. Numbers in each column indicate the number of individuals in each state. Different letters (a and b) above columns indicate statistically significant differences (*p* < 0.05, Fisher’s exact test with Bonferroni correction). (**B**) Effects of insecticides on gall growth. Means ± standard deviations are shown. Sample size is given in parentheses. Different letters (a and b) indicate statistically significant differences on the same day (*p* < 0.05, pairwise Wilcoxon rank sum test with Bonferroni correction). ns means no significant difference.

**Figure 3 insects-15-00063-f003:**
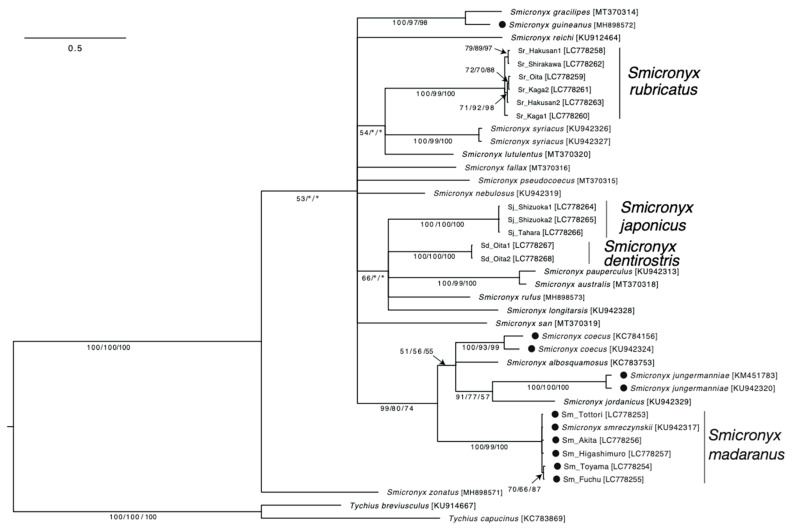
Molecular phylogeny and the gall-forming trait in the genus *Smicronyx*. The topology and branch lengths shown were obtained using Bayesian inference methods. The scale bar indicates 0.5 substitutions per site. The Bayesian posterior probabilities/the maximum likelihood bootstrap value/the maximum parsimony bootstrap value are shown below the branches to indicate the level of support for each node (only values ≥ 50% are shown), respectively. An asterisk (*) indicates a node that was not supported by ML or MP analyses. Sequences obtained in this study are in bold type. Labels of the sequences correspond to the codes in Appendix A. Filled circles in the taxa represent the gall-forming species. Sequence accession numbers are in brackets.

**Figure 4 insects-15-00063-f004:**
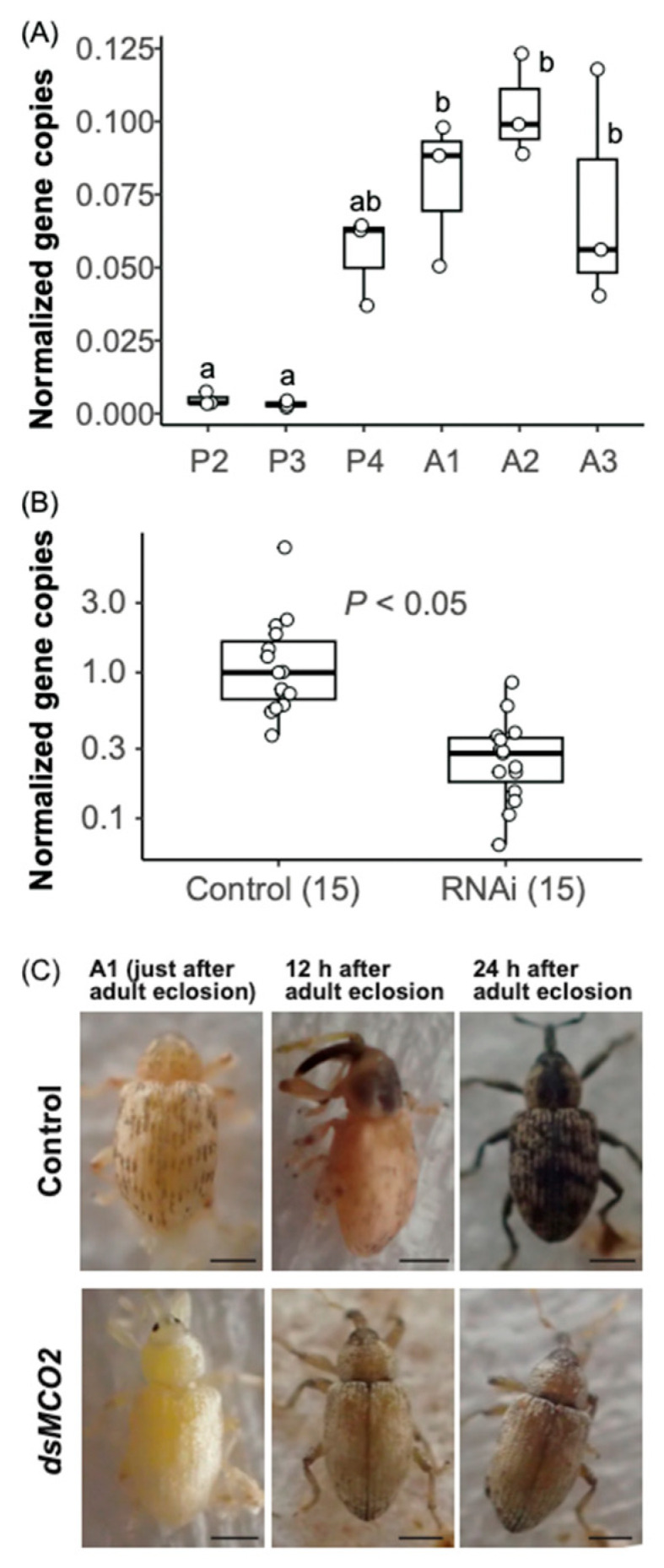
*SmMCO2* expression and body color melanization. (**A**) Variation in *SmMCO2* expression in different growth stages. Box plots show the distribution of the normalized *SmMCO2* expression level. Different letters (a and b) indicate statistically significant differences on the same day (*p* < 0.05, pairwise Wilcoxon rank sum test with Bonferroni correction). Growth stages (P2 to A3) correspond to Appendix A. (**B**) Effect of RNAi on the normalized *SmMCO2* expression level. The numbers in parentheses are sample sizes. The median expression level of the control group was designated as 1.0. Statistically significant differences were evaluated using Welch’s *t*-test. In (**A**,**B**), *y* axes indicate *SmMCO2* gene copies/an *EF1α* gene copy. Each open circle indicates the value per sample. (**C**) Effect of RNAi for *SmMCO2* on body color. Bars, 1 mm.

**Table 1 insects-15-00063-t001:** Effects of ovipositor sealing on gall formation.

Treatment	n ^a^	Egg-Laying Hole ^b^	Gall ^c^
Control	12	12.42 ± 5.14	8.67 ± 3.55
Sealed	12	4.17 ± 2.25 ***	0 ***

^a^ Number of individuals examined. ^b^ Number of egg-laying holes formed in the nodes per individual. ^c^ Number of galls formed by an individual. Statistical significance was calculated using Fisher’s exact test (***, *p* < 0.001).

## Data Availability

The original contributions presented in the study are included in the article/Appendix A, further inquiries can be directed to the corresponding author.

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
