# Peer review of "New Gall-Forming Insect Model, Smicronyx madaranus: Critical Stages for Gall Formation, Phylogeny, and Effectiveness of Gene Functional Analysis"

_insects, 2024, doi:10.3390/insects15010063_

Round 1

Reviewer 1 Report

Comments and Suggestions for Authors

Review of the manuscript: "New gall-forming insect model, Smicronyx mandaranus: critical stages for gall formation, phylogeny, and effectiveness of gene functional analysis".

The results presented in the manuscript are the part of the research on gall-formation mechanisms and this type of study is of great value. They bring us closer to answer the questions about the mechanisms of galls induction and formation. The manuscript is well-written, well-structured and clear. I think that the interpretations of the data to reach conclusions are fine. I have a few comments listed below:

1) the last part of title is: ‘…. and effectiveness of gene functional analysis’, later on in the subsection 3.5. the Authors state that: ‘However, functional analysis of the genes is necessary to prove causality, that they are responsible for the gall formation’ (lines 350-351) and in section Conclusion and prospective there is the statement: 'It is expected that the mechanisms of gall induction in adult females can be elucidated through a comprehensive functional analysis of candidate genes using RNAi'. The Authors proved that RNAi was effective in adult S. madaranus targeting the MCO2 gene, however they do not write anything if MCO2 gene can affect gall formation;

2) fig. 1B should be in the subsection 3.1.;

3) line 285: statement ‘escapade’ should be changed;

4) information in first paragraph (line 400-407) in the section Conclusion and perspectives, is a repetition of the Introduction, so it can be omitted or shortened.

Author Response

We are grateful for your important comments. We appreciate the time you have devoted to critically reviewing our manuscript.

Reviewer 2 Report

Comments and Suggestions for Authors

This paper is on an interesting topic, the gall-forming weevil Smicronyx madaranus.

1.       I think it is premature to conclude that “Phylogenetic analysis showed that most gall-forming weevils in the genus Smicronyx belonged to a single lineage.” (Lines 21-22) and “Phylogenetic analysis showed that most gall-forming weevils, including S. madaranus, belong to a single lineage, suggesting that gall- forming traits evolved in a specific lineage within Smicronyx.” (Lines 33-35). Only 4 gall formers on Cuscuta spp. and one gall-forming species on Striga spp. were part of the phylogenetic analysis.  Only 23 species of the more than 130 species of Smicronyx that have been described were part of the phylogenetic analysis. The discussion regarding the phylogenetic analysis should be expanded. There should be consideration of whether the species forms galls on Cuscuta spp. as opposed to other species of plants such as Striga spp. Only one of the species that was analyzed in this study (S. guineanus) forms galls on Striga spp. and it was not in the clade with the other gall formers that were analyzed that form galls on Cuscuta spp.  This should have been noted in the discussion. The fact that S. rubricatus feeds on Cuscuta but does not make galls, should be included in the analysis of the results. It would be interesting to see if the North American species S. sculpticollis that also form galls on Cuscuta belongs in the same lineage as the other 4 Cuscuta gall-forming species that were part of the analysis of this paper.  

2.       It is not particularly clear how the section dealing with the efficacy of RNAi as a functional gene analysis technique fits in with the other goals of the paper. I would suggest leaving this out of this particular paper.

3.       The premise that “that most insect galls are produced on woody plants and therefore cannot be experimentally validated in the laboratory” (lines 27-28 and 55) is not accurate for all parts of the world.  There are many gall-forming insects that induce galls on herbaceous plants.

4.      Line 273- 274: “These results suggest that gall induction is strongly related to eggs and substances” would be more accurately stated as “These results suggest that gall induction is strongly related to eggs and/or substances".

5.       Line 307: “This suggests that gall-formation mechanisms are diverse, even within weevil species.” should be reworded to “This suggests that gall-formation mechanisms are diverse, even among different weevil species.”

Comments on the Quality of English Language

The English is quite good.

Here are a few suggested changes:

1.       Line 65: “rostrum” should be “rostrums”

2.       The word “petri” is typically capitalized as Petri dishes are named after an individual named Petri.

3.       Line 88: “strain” should be “strains”

4.       Line 155, “laser blade” should presumably be “razor blade”??

5.       Line 259 “presented” should be “present”

Author Response

(The authors gave the same response as above.)
